# Thermosensitive Drug Delivery System SBA-15-PEI for Controlled Release of Nonsteroidal Anti-Inflammatory Drug Diclofenac Sodium Salt: A Comparative Study

**DOI:** 10.3390/ma14081880

**Published:** 2021-04-09

**Authors:** Lubos Zauska, Stefan Bova, Eva Benova, Jozef Bednarcik, Matej Balaz, Vladimir Zelenak, Virginie Hornebecq, Miroslav Almasi

**Affiliations:** 1Department of Inorganic Chemistry, Faculty of Science, P. J. Šafárik University, Moyzesova 11, SK-041 01 Košice, Slovakia; lubos.zauska@student.upjs.sk (L.Z.); evapopjakova@gmail.com (E.B.); vladimir.zelenak@upjs.sk (V.Z.); 2BovaChem s.r.o, Garbiarska 1919/14, SK-048 01 Rožňava, Slovakia; stefan.bova@gmail.com; 3Institute of Experimental Physics, Slovak Academy of Sciences, Watsonova 47, SK-040 01 Košice, Slovakia; jozef.bednarcik@upjs.sk; 4Institute of Geotechnics, Slovak Academy of Sciences, Watsonova 45, SK-040 01 Košice, Slovakia; balazm@saske.sk; 5Aix-Marseille University, CNRS, MADIREL, F-133 97 Marseille, France; virginie.hornebecq@univ-amu.fr

**Keywords:** mesoporous silica, surface modification, polyethylenimines, diclofenac sodium, temperature and pH, kinetic models

## Abstract

Mesoporous SBA-15 silica material was prepared by the sol–gel method and functionalized with thermosensitive polyethylenimine polymers with different molecular weight (g·mol^−1^): 800 (SBA-15(C)-800), 1300 (SBA-15(C)-1300) and 2000 (SBA-15(C)-2000). The nonsteroidal anti-inflammatory drug (NSAID) diclofenac sodium was selected as a model drug and encapsulated into the pores of prepared supports. Materials were characterized by the combination of infrared spectroscopy (IR), atomic force microscopy (AFM), transmission electron microscopy (TEM), photon cross-correlation spectroscopy (PCCS), nitrogen adsorption/desorption analysis, thermogravimetry (TG), differential scanning calorimetry (DSC) and small-angle X-ray diffraction (SA-XRD) experiments. The drug release from prepared matrixes was realized in two model media differing in pH, namely small intestine environment/simulated body fluid (pH = 7.4) and simulated gastric fluid (pH = 2), and at different temperatures, namely normal body temperature (T = 37 °C) and inflammatory temperature (T = 42 °C). The process of drug loading into the pores of prepared materials from the diclofenac sodium salt solutions with different concentrations and subsequent quantitative determination of released drugs was analyzed by UV-VIS spectroscopy. Analysis of prepared SBA-15 materials modified with polyethylenimines in solution showed a high ability to store large amounts of the drug, up to 230 wt.%. Experimental results showed their high drug release into the solution at pH = 7.4 for both temperatures, which is related to the high solubility of diclofenac sodium in a slightly alkaline environment. At pH = 2, a difference in drug release rate was observed between both temperatures. Indeed, at a higher temperature, the release rates and the amount of released drug were 2–3 times higher than those observed at a lower temperature. Different kinetic models were used to fit the obtained drug release data to determine the drug release rate and its release mechanism. Moreover, the drug release properties of prepared compounds were compared to a commercially available medicament under the same experimental conditions.

## 1. Introduction

Oncological and/or infectious diseases often cause inflammation in the human body [1]. Inflammation can have a local character or spread throughout the whole human body, accompanied by symptoms such as pain, fever and other complications. Anti-inflammatory medications such as nonsteroidal anti-inflammatory drugs (NSAIDs) and analgesics [2] are effective inflammatory suppressants. However, these medications are often invasive and addictive and can exhibit negative long-term side effects such as liver, kidney, heart and bone marrow damage. A method that has proven to be relatively significant is the use of a targeted drug delivery system (DDS) [3]. DDSs are biocompatible structures in which the drug is encapsulated and then released only at the targeted site is reached. Carriers can be inorganic materials, such as metal–organic frameworks (MOFs) [4,5], carbon nanotubes [6], fullerenes [7], zeolites [8] and silica [9], or organic materials, such as polymers [10], micelles [11] and dendrimers, containing different functional groups. It is also possible to make DDSs from proteins, lipids and polysaccharides [12], but the most common materials are combinations of mentioned structures [13]. 

In bioinorganic chemistry and pharmacology, mesoporous silica materials have found application due to their excellent biocompatibility, low cytotoxicity, uniform particle size, resistance and stability [14]. The surface of silica can be modified by various organic and inorganic groups that are called “plug” and that respond to external stimuli. The mentioned surface plug prevents the uncontrolled release of a drug from the porous structure of silica [15]. Different molecules have been studied as a plug, from the smallest binary compounds [16] to polymeric [17,18,19] and macrocyclic compounds [20,21]. All substances are sensitive to changes in external stimuli such as temperature [22,23], pH [24,25,26,27], UV radiation [28,29], magnetic field [30], electric current [27,31] and others [32]. 

Polyethyleneimines (PEIs) have interesting properties and have been shown to be pH- and temperature-sensitive polymers [33]. PEI molecules (linear or branched form) are widely studied and applied in DDSs as coated monolayers on inorganic nanoparticles, such as in the case of Fe_3_O_4_/Gd_2_O_3_ where cisplatin was used as a model drug [34], or bioorganic substrates such as cellulose for encapsulation of salicylate sodium [33]. PEI was also studied as a DDS for doxorubicin without any supportive substrate, which was assembled into spherical nanogel via cross-linking [35]. It can be possible to bind PEI molecules via disulfide bonds as an enzyme-sensitive DDS and entrap ceftriaxone sodium [36]. The versatility of PEI allows merging with other polymers such as polyethyleneglycol [37] and poly(N-isopropylacrylamide) [38] to synthesize multifunctional polymers for DDSs. According to database search and the best of our knowledge, silica modified with PEI (linear form) has only been studied in one article as a DDS for anticancer therapy [39]. In the mentioned study, mesoporous silica with ellipsoidal-shaped grains with a particle size of 40 to 120 nm was used as a support. The surface of support was modified with linear PEI molecules and folic acid. Curcumin as the drug was encapsulated into the pores of surface-modified material with a maximal storage capacity of about 80 wt.%. The drug release was assessed at different pH values, and the amount of drug released after 120 h was 54.6 wt.% at pH = 5.4, 18.52 wt.% at pH = 6.8 and 8.87 wt.% at pH = 7.4. The uniqueness of the presented study lies in the investigation of thermosensitive properties of PEI polymers, which were first studied on the porous silica material SBA-15.

In our study, the drug diclofenac sodium, which belongs to the group of nonsteroidal anti-inflammatory drugs (NSAIDs) was used. Diclofenac sodium, or 2-[2-(2,6-dichloroanilino)phenyl]acetate sodium (according to IUPAC, see Figure 1) is used in non-rheumatic, rheumatic and inflammatory diseases. It acts as a mediator in the inhibition of prostaglandin synthesis in cyclooxygenase. In an in vivo study, diclofenac did not inhibit phospholipase A2 at a high concentration that controlled arachidonic acid synthesis and had low to no effect on 5- and 15-lipoxygenases. Nevertheless, the concentrations of products derived from the lipoxygenase pathway were reduced precisely due to the high concentrations of diclofenac in vivo and exclusively in leukocytes. This effect is likely to be caused by the availability of intracellular arachidonic acid [40].

In the present study, thermosensitive DDSs based on mesoporous silica SBA-15 grafted with branched polyethyleneimines (PEIs) with different molecular weights (*M_w_* = 800, 1000 and 1200 g·mol^−1^) were prepared, characterized and studied as supports for nonsteroidal anti-inflammatory drug diclofenac sodium (DIC). Drug adsorption into the prepared support performed in solution revealed high DIC storage capacity up to 230 wt.%. The in vitro release properties were assessed in two model media differing in pH, namely small intestine environment/simulated body fluid (pH = 7.4) and simulated gastric juice (pH = 2), and at different temperatures, namely T = 37 °C simulating normal human body temperature and T = 42 °C representing inflammatory temperature. These two temperatures were chosen as external stimuli for the controlled release of diclofenac sodium. Indeed, at a lower temperature (T = 37 °C), the PEI molecules form hydrogen bonds with each other and the entry into the pores of the material should be blocked (closed conformation). At a higher temperature (T = 42 °C), the hydrogen bonding system should be temporarily disrupted, the pores of the carriers should be opened, accelerating the release of the drug (open conformation). In addition, the data obtained from drug release studies were fitted using different kinetic models to calculate the release rate and release mechanism of diclofenac sodium from porous matrixes.

## 2. Materials and Methods

### 2.1. Used Chemicals

All chemicals used in this study were purchased from Sigma-Aldrich Company (Saint-Louise, MO, USA) in the highest possible purity. Pluronic-123 (*M_n_* = 5800); (3-chloropropyl)triethoxysilane (95%); tetraethyl orthosilicate, reagent grade (98%); and branched polyethylenimine (*M_w_* = 800, 1300 and 2000 g mol^−1^, corresponding to 600, 1200 and 1800 monomeric units (*M_n_*), respectively) were used for SBA-15 synthesis and surface modification. Diclofenac sodium salt (sodium 2-[(2,6-dichlorophenyl)amino]benzeneacetate, see Figure 1), >98%, was purchased from Sigma-Aldrich (St. Louis, MO, USA) and used as a model drug. In comparative experiments, Diclofenac Duo PharmaSwiss was used as a reference for controlled drug release.

### 2.2. Preparation of SBA-15

First, 13.1 g of surfactant P-123 was dissolved in a mixture of 382 mL of hydrochloric acid (2M solution) and 98 mL of deionized water. The reaction mixture was stirred for 48 h at 35 °C and 580 rpm. After the dissolution of Pluronic, 28 mL of TEOS was added dropwise 1 mL/min into the mixture under stirring. The whole reaction was stirred at 580 rpm and temperature 35 °C for 24 h. The reaction mixture was then placed in an oven which was preheated at 80 ℃, and the material was aged for 24 h. After the aging process, powder material was filtered off and the solid residue was washed several times with deionized water and dried in the stream of air (yield 17.3 g). The as-synthesized sample is denoted SBA-15(AS) in the following.

### 2.3. Extraction/Calcination Process

After the material drying procedure, surfactant molecules located in pores were removed by the combination of extraction and calcination processes. The first step was an extraction with dry toluene to purify the silica from contamination of impurities that occurred during the synthesis. The second step was sample extraction with HCl (2M) to remove the surfactant from the pores. The third step of the extraction was performed with tetrahydrofuran to remove the remaining surfactant molecules and other impurities. After extraction processes, the sample was dried in an oven at 70 °C and further calcined. The calcination process was performed in an oven with the following temperature program: First, the sample was slowly heated, with a heating rate of 0.5 °C·min^−1^, to 150 °C for 2 h. Slow heating was chosen to remove residual solvents from the mesopores of the material. The next step was thermolysis of the residual surfactant by heating to 600 °C with a slow heating rate of 1.25 °C·min^−1^. After heating the material at 600 °C for 8 h, the material was slowly cooled down to ambient temperature. The weight of the calcined material was 3.585 g, and the sample after the calcination process was denoted as SBA-15(C).

### 2.4. Surface Grafting and Modification

The prepared porous material SBA-15(C) was subjected to surface grafting (see Figure 2) with 3-(chloropropyl)-trimethoxysilane (CPTES). Three grams of SBA-15(C) was first dispersed in 150 mL of dry toluene, and then 9 mL of 3-(chloropropyl)-trimethoxysilane was added dropwise. After the addition of the silane derivative, the reaction mixture was heated for 24 h under reflux at 130 °C with a stirring speed of 500 rpm. Subsequently, the reaction mixture was cooled down to ambient temperature, filtered off, washed with dry toluene and dried in an oven at 50 °C overnight. To eliminate the unreacted species of 3-(chloropropyl)-trimethoxysilane from grafted material, 8 h extraction with dry toluene was performed. The mesoporous material after the grafting process with chloropropyl groups was designated as SBA-15(C)-Cl (yield 3.5 g).

Chloropropyl-grafted material (SBA-15(C)-Cl) was prepared as a precursor for post-synthetic modification of the surface with polyethylenimines (PEIs) with different molecular weights: 800, 1300 and 2000 g·mol^−1^. For PEI binding, the surface must contain free chloropropyl function groups that can react with primary or secondary amine groups of polyethylenimines by a condensation reaction to form a covalent C–N bond and release a HCl molecule (see Figure 2). In this way, 0.8 g of SBA-15(C)-Cl was dispersed in 25 mL of dry toluene, and then 3.3 g of corresponding PEI (800, 1300 and 2000 g mol^−1^) was added and synthesis was performed under reflux at 130 °C for 24 h. Unreacted PEI molecules were removed from the pores of the prepared materials by extraction with 2M hydrochloric acid solution and then dried in an oven at 70 °C overnight. Materials after surface modification with polyethylenimines were designated as SBA-15(C)-800 (yield 3.30 g), SBA-15(C)-1300 (yield 3.47 g) and SBA-15(C)-2000 (yield 3.64 g).

### 2.5. Drug Encapsulation

Modified samples SBA-15(C)-800, SBA-15(C)-1300 and SBA-15(C)-2000 and calcined material SBA-15(C) were used as carriers for diclofenac sodium salt. The affinity of the drug to the surface of carriers was assessed using methanolic solutions of diclofenac sodium with different concentrations: 0.1, 0.05, 0.025, 0.0125 and 0.00625 mol·dm^−3^. Methanol was used as a solvent due to the high solubility of diclofenac sodium salt (35 mg·cm^−3^). In general, 10 mg of samples SBA-15(C)-PEI-X (X = 800, 1300 and 2000)/SBA-15(C) were dispersed in 1 mL of drug solution in a plastic vial. Suspensions were stirred at 40 rpm (rotator) for 24 h at 42 °C (open conformation) while the evaporation of methanol was prevented. The suspensions were then centrifugated at 6000 rpm for 10 min; after centrifugation, the supernatants were separated, and the amount of adsorbed drugs was determined by UV-VIS spectroscopy, based on the calibration curve of DIC in methanol (see Appendix A).

### 2.6. Drug Release

Drug-loaded samples (~3 mg) were packed in a semipermeable membrane VISKING and inserted into 50 mL polypropylene vials covered with caps. The vials were filled with 50 mL of saline solutions with pH = 2 and pH = 7.4 and transferred into a preheated oven at 37 °C (closed conformation) or 42 °C (open conformation), and mixtures were gently stirred on a magnetic stirrer. In general, the release amount of the drug was analyzed at selected time intervals: 0.5, 1.5, 3.5, 5.5, 7.5, 9.5 and 24 h. 

The amount of the drug released was determined using UV-VIS spectroscopy. Before determining the concentration of DIC, calibration curves were constructed based on solutions (saline at pH = 7.4/2/5.5) with various concentrations of diclofenac sodium (see Appendix A). The correlation coefficient *r*^2^ = 0.9999 for saline solution at pH = 7.4, *r*^2^ = 0.9966 for saline solution at pH = 2 and *r*^2^ = 0.9999 for saline solution at pH = 5.5 confirmed the linearity of calibration curves. pH = 5.5 was used in drug release experiments performed under dynamic conditions.

### 2.7. Drug Solubility

The diclofenac sodium solubility was studied at 37 and 42 °C and pH 2 and 7.4. One hundred milligrams of drug was dispersed in 10 mL of saline solution with different pH values in glass vials. Vials were sealed and sonicated for 3 min. After sonification, vials were put in the preheated ovens at 37 and 42 °C. The prepared suspensions were stirred under selected conditions (1. 37 °C, pH = 2; 2. 37 °C, pH = 7.4; 3. 42 °C, pH = 2; 4. 42 °C, pH = 7.4) for 24 h. After that, the remaining undissolved drug was filtered with polytetrafluoroethylene (PTFE) micro filter with 0.2 µm porosity and 20 mm diameter. Subsequently, 100 μL of supernatant from the individual solutions was diluted in water in a 1:1000 ratio (drug solution/saline solution) and analyzed by UV-VIS spectroscopy (Analytik Jena GmbH, Jena, Germany). The amount of drug in the samples was determined based on absorbance using calibration curves.

### 2.8. Characterization

Infrared spectra of prepared materials were measured by FT-IR spectroscopy on Nicolet 6700 from Thermo Scientific (Thermo Scientific, Waltham, MA, USA) using KBr technique in the wavelength range of 4000–400 cm^−1^. Samples for IR analysis were prepared in the form of KBr pellets with KBr/sample mass ratio 100:1 or 100:5. For selected samples, a higher mass ratio was chosen for better visibility of characteristic absorption bands. Before IR measurements, potassium bromide was dried at 700 °C for 4 h in an oven to remove water and freely cooled in a desiccator. All IR spectra of prepared materials were recorded by collecting 64 scans with a resolution of 4 cm^−1^ for a single spectrum at ambient temperature. 

Nitrogen adsorption/desorption isotherms were measured at −196 °C and experiments were carried out using Nova 1200e from Quantachrome Instruments (Quantachrome, Miami, FL, USA) and ASAP 2020 Micromeritics apparatus (Micromeritics, Norcross, GA, USA). Before nitrogen adsorption/desorption measurements, samples were outgassed at different temperatures (150 °C for SBA-15(C) and extracted samples, 60 °C for surface-modified materials). The Brunauer–Emmett–Teller (BET) specific surface area (*S_BET_*) of each sample was calculated using nitrogen adsorption data in a p/p_0_ = 0.05–0.20. The textural properties such as pore size diameter (*d*) and pore volume (*V_p_*) and were evaluated using the Barrett–Joyner–Halenda (BJH) model from the nitrogen desorption isotherm. 

The shape and surface morphology of prepared materials were analyzed using atomic force microscopy (AFM) on a Solver PRO instrument from NT-MDT Spectrum Instruments (NT-MDT, Moscow, Russia). The measurements were performed in contact and semi-contact mode on samples attached to a surface-treated mica plate with polylysine.

Transmission electron microscopy was carried out on a microscope JEOL 2000FX (JEOL, Pleasanton, FA, USA). Prepared materials were ground and dispersed in ethanol. The material’s suspension was dripped onto a carbon grid and freely air-dried.

The particle size distribution in the nano-range was determined by a photon cross-correlation spectroscopy, which was carried out on a Nanophox particle size analyzer (Sympatec, Clausthal-Zellerfeld, Germany). A small amount of powder product was dispersed in the deionized water and ultrasonicated for 15 min. After that, the obtained suspensions were left standing for 10 min, and subsequently, part of the top fraction containing a significant amount of fine particles was transferred into the measurement cuvette and diluted with distilled water to obtain a suitable concentration for the experiment. A dispersant refractive index of 1.33 was used for the analysis, and the measurements were repeated three times for each sample. The polydispersity index (PDI) was calculated by dividing the square of the standard deviation of the corresponding peak by the square of the value of its central position on the *x*-axis.

Small-angle X-ray diffraction (SA-XRD) experiments (Rigaku, Tokyo, Japan) were performed on a Rigaku Ultima IV multipurpose diffractometer in transmission geometry using CuKα radiation (*λ* = 1.54056 Å). The thickness of the sample during the measurement was 2 mm, which was achieved by placing the sample in a metal holder, which was sealed on both sides with Kapton tape. SA-XRD measurements were realized by 2*θ* continuous scan at 0.2° min^−1^ in the 2*θ* range from 0.1 to 3°, and scattered photons from the samples were detected every 0.02° using a sodium iodide scintillation detector.

TGA Q500 apparatus from TA Instruments (TA Instruments, New Castle, DE, USA) was used for determining the thermal properties of prepared materials. The thermogravimetric analysis was performed in the temperature range of 30–800 °C with a heating rate of 10 °C min^−1^ in an air atmosphere with a flow rate of 60 cm^3^·min^−1^ using platinum crucibles.

The volume phase transition behavior of PEI-modified samples was investigated by differential scanning calorimetry (DSC) on a DSC Q2000 apparatus from TA Instruments (TA Instruments, New Castle, DE, USA). The samples were heated with a heating rate of 1 °C min^−1^ in a temperature range of 10–60 °C and nitrogen atmosphere.

UV-VIS spectroscopy was performed on a Specord 250 UV-VIS spectrometer from Analytik Jena AG (Analytik Jena GmbH, Jena, Germany) in a wavelength range of 240–350 nm. UV-VIS spectroscopy was used to determine the amount of adsorbed/released drug.

## 3. Results and Discussion

### 3.1. Synthesis of SBA-15 and Surface Modification

The sol–gel process was used for the synthesis of SBA-15 by hydrolysis and polycondensation reactions of TEOS in water using Pluronic-123 as surfactant and HCl as a catalyst [41]. Sol is a low viscosity liquid that can be transformed into a solid form. Over time, the colloidal and condensed silica are combined into a 3D structure to form a mesh. The gel form and its physical properties depend mainly on the particle size. When the gel is forming, the viscosity increases the strength and integrity of the silica material. The structure and textural properties of the gel depend on the time of its formation. Other processes such as aging, drying, stabilization and thickening depend on the structure of the gel [42]. Achieving the largest silica surface area, pore size and pore volume depend on the removal of surfactant molecules. Calcination of silica nanoparticles at high temperature is a fast and efficient way to remove surfactant from pores. However, the calcination conditions must be chosen carefully because this process can perforate the silica pores. The as-synthesized silica contains not only surfactant molecules but also molecules of solvent in the pores. With rapid calcination, there is a risk that the solvent molecules will start to evaporate quickly and perforate the pores. Surfactant extraction is a slower but more gentle removal process that, on the other hand, does not lead to the complete removal of surfactant molecules. For the mentioned reasons, we have chosen in our work a combination of extraction and calcination of SBA-15(AS) material. The extraction was performed using solvents with different polarity (toluene, THF and 2M HCl solution), and then the material was calcined by slow heating at 600 °C. For modification with organic compounds, the surface was treated by silylation reaction with alkoxysilanes and similar silane compounds. Mentioned surface grafting is suitable for the subsequent attachment of organic compounds with different functional groups. The surface of SBA-15(C), which contains free hydroxyl groups, was subsequently grafted with 3-(chloropropyl)-trimethoxysilane molecules in dry toluene. This step was necessary for the formation of the covalent bond between the surface of SBA-15(C) and polyethylenimine molecules, via a condensation reaction. PEI molecules with different molecular weights (800, 1300 and 2000 g·mol^−1^) were used for surface modification. The materials prepared by the described processes were subsequently used as carriers for the anti-inflammatory drug diclofenac sodium and tested as thermosensitive DDSs.

### 3.2. Infrared Spectroscopy

#### 3.2.1. SBA-15 and Surface-Modified Samples

Infrared spectra of prepared materials are presented in Figure 3, and the assignment of characteristic bands is summarized in Table 1. The infrared spectrum of SBA-15(C) (see Figure 3a) shows a broad band at 3437 cm^−1^ corresponding to stretching *ν*(OH) vibration of silanol groups located on the surface of silica (Si–OH) and physisorbed water molecules. Bending vibrations *δ*(OH) of mentioned groups was found at 1638 cm^−1^. The presence of silica framework is reflected by broad and intense bands at 1088 and 809 cm^−1^, which can be attributed to the asymmetric (*ν_as_*(SiOSi)) and symmetric (*ν_s_*(SiOSi)) stretching vibrations, respectively. Bending vibrations of δ(SiOSi) groups were detected at 462 cm^−1^.

The IR spectrum of SBA-15(C)-Cl after grafting with 3-(chloropropyl)-trimethoxysilane is shown in Figure 3b. Bands observed at the wavelengths of 2981, 2931 and 2900 cm^−1^ are attributed to the valence vibrations of *ν*(CH_2_) groups of propyl chain. Deformation vibration (*δ*(CH_2_)) of mentioned groups appeared at 1453 cm^−1^. 

All IR spectra (see Figure 3c) of surface-modified samples with polyethylenimines (SBA-15(C)-PEI-X (X = 800, 1300 and 2000)) contain characteristic bands in the wavenumber range of 3360–3251 cm^−1^ due to stretching vibrations of primary/secondary amines (*ν*(NH_2_)/*ν*(NH)) of PEI molecules. In the IR spectra, deformation bands *δ*(NH_2_) in the range from 1602 to 1603 cm^−1^ are present, which also confirm the presence of amine groups. Moreover, the presence of PEI molecules is also evidenced by the presence of two weak stretching vibrations of aliphatic *ν*(CH) groups of ethylene bridges located below 3000 cm^−1^ (see Table 1, Figure 3c).

#### 3.2.2. Diclofenac Sodium and Drug-Loaded Samples

The structure of diclofenac sodium is presented in Figure 1, and the following bands were observed in its IR spectrum (see Appendix A): a weak band at 3080 cm^−1^ belonging to the stretching vibration of aromatic CH groups of phenyl scaffold; sharp, medium-intensity bands at 1496, 1573 and 1604 cm^−1^ corresponding to the valence vibrations of the conjugated system of C=C bonds in the phenyl ring; a low-intensity, sharp band at about 717 cm^−1^ related to the deformation vibrations for monosubstituted phenyl ring; and three low-intensity, sharp bands at 767, 951 and 1170 cm^−1^ corresponding to the deformation vibrations for 1,2,3-trisubstituted phenyl ring. Moreover, in the IR spectrum of diclofenac sodium salt, a strong and sharp band was found at 747 cm^−1^, which could be assigned to the vibration of the C–Cl bond. The presence of a secondary amine group is obvious from the weak and broad band at 3386 cm^−1^ (*ν*(NH)), and its deformation vibration *δ*(NH) was detected at 1556 cm^−1^. Other characteristic vibrations in the IR spectrum of diclofenac sodium are symmetric and asymmetric vibrations of the carboxylate group (*ν*(COO^−^)), located at 1400 and 1550 cm^−1^, respectively.

The IR spectra of samples with encapsulated DIC are shown in Figure 3d, and characteristic bands are listed in Table 2. The presence of diclofenac sodium salt encapsulated in prepared materials is evidenced by the presence of two sharp, medium-intensity bands at 1555 and 1580 cm^−1^, which correspond to the valence vibrations of the aromatic ring (*ν*(C=C)_ar_). Typical vibrations of carboxylate group *ν*(COO^−^)_s_ and *ν*(COO^−^)_as_ were found at about 1450 and 1510 cm^−1^. Another low-intensity, sharp band found at 747 cm^−1^ corresponds to the valence vibration of the C–Cl bond. 

### 3.3. Nitrogen Adsorption/Desorption Measurements

The specific surface area (*S_BET_*), pore size (*d*) and pore volume (*V_p_*) of prepared materials were determined by nitrogen adsorption/desorption measurement at −196 °C. Obtained adsorption/desorption isotherms are depicted in Figure 4, and calculated textural properties are listed in Table 3. 

The extraction/calcination process of as-synthesized SBA-15 was monitored by nitrogen adsorption. The stepwise removal of surfactant from SBA-15(AS) by extraction with toluene (pink curve), hydrochloric acid (black curve), tetrahydrofuran (blue curve) and final calcination (red curve) is depicted in Figure 4a, and calculated textural parameters are summarized in Table 3. A detailed synthetic procedure for surfactant removal is given in Section 2.3. As can be seen from Figure 4a, the use of different polar solvents for the extraction procedure led to the gradual removal of surfactant, resulting, for samples, in an increase in both adsorbed nitrogen volume and associated surface area. The toluene-extracted sample showed a very low surface area of 14 m^2^ g^−1^. After extraction of the material with hydrochloric acid, there was a high increase in the *S_BET_* value to 547 m^2^ g^−1^. Subsequent extraction with tetrahydrofuran allowed the pore opening of SBA-15 to continue (744 m^2^·g^−1^), and complete removal of the surfactant occurred after calcination of the material at 600 °C. The specific surface area of calcined material SBA-15(C) was 814 m^2^ g^−1^, which was further used for the PEI grafting process. 

The opposite trend, i.e., the decrease in specific surface area, was observed for materials after the grafting process using 3-(chloropropyl)-trimethoxysilane and PEI molecules with different molecular weights. After grafting the surface with chloropropyl groups, a decrease in *S_BET_* was observed from 814 m^2^·g^−1^ (SBA-15(C)) to 470 m^2^·g^−1^ (SBA-15(C)-Cl). The textural properties of materials modified with PEI molecules showed that the polymers were predominantly grafted to the external surface of grains and at the entrances of the pores (and not in pores) as the specific surface areas were further reduced with PEI molecules, reaching 206 m^2^·g^−1^ (SBA-15(C)-800), 184 m^2^·g^−1^ (SBA-15(C)-1300) and 144 m^2^·g^−1^ (SBA-15(C)-2000), and because the pore size decreased only minimally compared to the sample SBA-15(C)-Cl (see Table 3). Thus, PEI molecules predominantly occupy the particles’ outer surface and partially prevent the effective diffusion of nitrogen into the internal pores. This fact was also confirmed by AFM spectroscopy, which clearly showed the PEI overlapping the entrances to the pores (see Section 3.4).

### 3.4. Surface Morphology and Particle Size 

The atomic force microscopy (AFM) scan images and selected TEM micrographs of calcined SBA-15(C) material, propylchloride-grafted sample SBA-15(C)-Cl and polyethylenimine-modified material SBA-15(C)-800 are depicted in Figure 5. AFM was used to determine the surface morphology, shape and size of the pores in prepared materials. The AFM images (Figure 5a,b) and TEM micrographs (Figure 5c,d) display the uniform tubular character of pores with a size of 6.5 × 7 nm^2^, which are characteristic of mesoporous silica of the SBA-15 type. As can be seen from Figure 5a–d, the pores have a regular hexagonal structure, are placed next to each other and have an almost uniform structure. Figure 5e shows an AFM scan of sample SBA-15(C)-Cl, which indicates a porous surface with a “rougher” structure compared to the surface of calcined SBA-15(C) material (Figure 5a). The surface roughness confirms the presence of grafted chloropropyl units on the surface. The AFM scan images of SBA-15(C)-800 are presented in Figure 5f. The sample has the same roughened surface as in the case of SBA-15(C)-Cl, with the difference that the relief of the surface is more articulated than that seen in the image in Figure 5e. Taken together, these results show that the surface is coated with polyethylenimine, which forms a thin film.

Small-angle X-ray diffraction measurement based on scattered X-rays is an important technique in the analysis of condensate matter. In the case of porous silica materials, SA-XRD measurement provides important information about the pore organization in terms of symmetry and the unit cell size.

Measured SA-XRD patterns for prepared materials studied in this work are displayed in Figure 6. The SA-XRD patterns contain typical diffraction peaks associated with the presence of a regular hexagonal arrangement structure in the materials. The SA-XRD pattern of the calcined sample SBA-15(C) (see black line in Figure 6) shows the characteristic three well-defined reflections located at 1.019, 1.728 and 2.094° that could be indexed as (100), (110) and (200) reflections, respectively. In functionalized materials with PEI molecules and samples after DIC encapsulation, the decrease in intensity of (110) and (200) reflections can be seen, while the first intense peak corresponding to (100) reflection is visible in all SA-XRD patterns. In general, the distance between the specific planes and lattice constants can be calculated based on peak position. Based on 2*θ* angle values, *hkl* indexes and the quadratic form of the Bragg equation for the *p6mm* hexagonal symmetry (Equation (1)), the unit cell parameter *a* for all prepared materials was calculated.
(1)$$a=\lambda /\sqrt{3}\sin\theta$$

Calculated values revealed that unit cell parameter (*a*) is about 100.5 Å for all samples (see Table 4). Findings from SA-XRD measurements confirm a hexagonal pore structure with a *p6mm* point group symmetry of synthesized carriers and confirm the porous framework’s stability after the surface grafting process and drug loading. 

Particle size distribution of surface-modified and drug-loaded samples was determined by a photon cross-correlation spectroscopy (PCCS), and obtained results are presented in Figure 7. The first view of particle size distribution curves shown in Figure 7 shows that the vast majority of the particles are in the submicron range. The particle size is influenced by both the presence of diclofenac sodium and by the molecular weight of the PEI polymers used in the grafting process.

In diclofenac-free samples (Figure 7a), the shift of particle size to the smaller one can be observed with the increase in the number of monomeric units/molecular weight of the polymer. This might be caused by the repulsion of its hydrophobic groups, as larger functional groups are associated with greater repulsion forces, which lead to smaller particles at the end. Moreover, a change in modality can be observed. Initially, the bimodal particle size distribution with one maximum located at around 370 nm and the absolute maximum at around 550 nm can be observed. With increasing the PEI molecular weight, the contribution of smaller particles increases. Namely, for the SBA-15(C)-800 sample, the bimodal size distribution with the smaller fraction being slightly more abundant was evidenced. Further increasing the PEI molecular weight leads to almost complete eradication of the larger fraction of particles and reduction of particle size. Namely, for the SBA-15(C)-2000 sample, almost unimodal distribution with an absolute maximum at the particle size of 287 nm was evidenced.

For the samples containing the therapeutics, the trend seems to be the other way around; i.e., with the increase in PEI molecular weight, the particle size distribution changes from unimodal to polymodal. Namely, for the PEI-free sample, the unimodal distribution with the maximum around 240 nm was evidenced. For this sample, the polydispersity index was calculated to be 0.096, which means that the presented data are relevant [43]. Introducing PEI and increasing its molecular weight leads to a change into bimodal (SBA-15(C)-800), trimodal (SBA-15(C)-1300) and polymodal (SBA-15(C)-2000) particle size distribution. The observed effect might be connected with the interaction of PEI with the therapeutic or with the efficient elimination of diclofenac sodium from the structure of the porous material.

To show the above-discussed results in a simpler way, the relationship between the molecular weight of PEI polymer and average particle size x_50_ (50% of the particle diameters are smaller than this value) is shown in Figure 8.

The constant decrease in x_50_ value from 535 nm for the initial SBA-15(C) to 295 nm detected for SBA-15(C)-2000 sample is in accordance with the discussion above. The addition of diclofenac sodium changed the particle size distribution. The introduction of therapeutics showed an opposite trend; namely, x_50_ values equal to 237 and 500 nm for the PEI-free sample and the sample containing PEI with molecular weight 1300 g·mol^−1^, respectively, were found. The decrease in x_50_ value in the case of SBA-15(C)-2000/DIC sample is not relevant, as the polymodal distribution was evidenced in this case and it is also highly probable that a significant amount of micrometer-scale particles sedimenting before the PCCS measurement was present in this case.

### 3.5. Thermogravimetric Analysis and Calorimetry

The thermal stability of calcined SBA-15(C), samples after surface grafting process and samples after diclofenac sodium encapsulation was studied by thermogravimetric (TG) analysis. The obtained TG curves measured in 150–800 °C temperature range are presented in Figure 9, and calculated parameters are summarized in Table 5 and Table 6. Thermoanalytical curves and sample weight loss values were normalized at 150 °C [44], and removed weight losses were redistributed between the weight loss corresponding to the release of organic part and the residual mass. From TG results, the amount of PEI molecules bonded on the surface of supports and amounts of loaded diclofenac sodium in prepared materials were calculated.

The thermogravimetric curve of the material SBA-15(C) is displayed in Figure 9a. The TG curve indicates successful removal of surfactant from SBA-15(AS) pores after extraction and calcination processes as no significant weight losses were observed up to 600 °C. Above 600 °C, 0.25 wt.% weight loss on the TG curve was observed, which was associated with the silanol groups’ dehydroxylation and condensation on the sample surface. This value was taken into account as a reference in the calculation of desired parameters for functionalized samples. TG curve of the SBA-15(C)-Cl (Figure 9a) displayed a total weight loss of 6.01 wt.% corresponding to 58.8 mg of propylchloride groups grafted per 1 g of material (see Table 5). The thermogravimetric curves of samples SBA-15(C)-800, SBA-15(C)-1300 and SBA-15(C)-2000 are shown in Figure 9b. From the results of TG measurements, it was possible to calculate the amount of PEI units present in the prepared materials. Determined weight losses were 9.80 wt.% (92.6 mg·g^−1^) for SBA-15(C)-800, 14.95 wt.% (146.5 mg·g^−1^) for SBA-15(C)-1300 and 24.02 wt.% (237.5 mg·g^−1^) for SBA-15(C)-2000 and correspond to 0.116, 0.113 and 0.118 mmol g^−1^, respectively. Although PEI compounds with different numbers of monomer units (molecular weight) were used, it is evident from the obtained results that the quantity of PEI was approximately the same and did not depend on the bulk of PEI molecules (see the amount of organic part in Table 5). 

The TG analysis was also used to determine the weight of encapsulated DIC molecules in the PEI-modified/unmodified samples. Before the discussion of obtained results, the thermal behavior of diclofenac sodium is described. From Appendix A, it is evident that diclofenac sodium salt is thermally stable after heating to 250 °C. In the temperature range of 250–680 °C, the organic part’s thermolysis takes place in three different steps associated with a total weight loss of 83.7 wt.% (calculated mass loss 83.4 wt.%). The decomposition stops at 680 °C, and as the final decomposition product, Na_2_CO_3_ was identified (residual mass: observed 16.3 wt.%, calculated 16.6 wt.%). TG curves of diclofenac sodium loaded samples are depicted in Figure 9c, and calculated amounts of DIC are summarized in Table 6. The weight of DIC molecules was calculated from the difference between weight losses of PEI surface-modified materials and weight losses in samples with loaded drug. Moreover, the TG results obtained for pure diclofenac sodium were incorporated and used to calculate the weight losses corresponding to the DIC and residual masses in drug-loaded samples. As can be seen from Table 6, the encapsulated quantity of DIC can be sorted in following order (see Table 6): SBA-15-800/DIC (205.6 mg·g^−1^) > SBA-15/DIC (191.1 mg·g^−1^) > SBA-15-1300/DIC (178.7 mg·g^−1^) > SBA-15-2000/DIC (141.8 mg·g^−1^). It should be noted that the calculated values do not provide realistic information on the effect of PEI surface modification on the affinity of DIC to the surface of carriers. To obtain this information, it is also necessary to take into account the textural properties, especially the surface area of the materials. Materials can store different amounts of the drug in their porous structure because of different surface areas/pore volumes (see Table 3 above), and the quantity of loaded diclofenac sodium should increase with increasing *S_BET_* area. For this reason, the stored DIC amounts in milligrams per 1 g of the support were calculated as the weight of diclofenac sodium in milligrams per square meter of the carrier, and obtained results can be arranged in the following order: SBA-15(C)-2000/DIC (0.985 mg·m^−2^) ≈ SBA-15(C)-1300/DIC (0.971 mg·m^−2^) ≈ SBA-15(C)-800/DIC (0.998 mg·m^−2^) > SBA-15(C)/DIC (0.235 mg·m^−2^) (see Table 6). From the obtained values, it can be concluded that the affinity of diclofenac sodium is similar for all PEI-modified materials, due to the formation of electrostatic interactions or intermolecular hydrogen bonds between primary/secondary groups of bonded molecules and diclofenac sodium molecules. PEI-modified samples are able to encapsulate four times more amount of drug compared to SBA-15(C).

The lower critical solution temperature (LCST) of an aqueous solution of PEI-modified samples was investigated by DSC measurements. Figure 9d clearly confirms LCST transition of polyethyleneimine molecules in SBA-15(C)-800, SBA-15(C)-1300 and SBA-15(C)-2000 samples by the presence of endothermic peaks with a maximum at 40.2, 41.1 and 41.7 °C, respectively. Because the thermal effects are accompanied by low intensity [45,46], it can be assumed that there is only a partial change in the conformation of the PEI molecules at the observed temperatures.

### 3.6. Drug Adsorption Properties 

The drug adsorption properties of prepared materials were monitored in solution by UV-VIS spectroscopy based on a calibration curve of DIC in methanol (see Appendix A). From obtained UV-VIS spectra, diclofenac sodium contains a characteristic absorption band at 280 nm, which corresponds to an electron transition *n* → *π**. Diclofenac sodium was loaded into the supports by the impregnation method using the methanolic drug solutions with different initial concentrations: 6.25 × 10^−3^, 1.25 × 10^−2^, 2.5 × 10^−2^, 5 × 10^−2^ and 10^−1^ mol.dm^−3^. The determination of the supernatant concentration after adsorption, defined as the equilibrium concentration, allows the calculation of the amount adsorbed, *Q_ADS_*, by the following equation: (2)$$Q_{ADS}=\;\frac{V(C_{i}-C_{eq})}{m_{s}}$$
where *V* is the volume of the drug solution (dm^3^), *C_i_* (mol·dm^−3^) is the initial concentration, *C_eq_* (mol·dm^−3^) is the equilibrium concentration and *m_s_* is the mass (g) of adsorbent used.

Then, the adsorption isotherms of DIC onto SBA-15(C)-800, SBA-15(C)-1300 and SBA-15(C)-2000 porous solids at 42 °C were determined, and obtained results are presented in Figure 10. A similar shape of isotherm was found for the three solids, corresponding to an L-type one [47]. L-type adsorption isotherm is frequently found in the case of energetic adsorption sites with a narrow and homogeneous distribution. 

Very similar values of DIC adsorbed amounts are found as isotherms are nearly superimposed. Furthermore, at low concentrations the slope of adsorption isotherms is very similar for the three solids and weak, evidencing a weak affinity of DIC for each solid. This is a very good result considering that DIC should be released afterward. Thus, it seems that there is no impact of the polymer chain length on the DIC adsorption properties. As shown in Figure 11, it is apparent that the prepared carriers can store high amounts of the drug: 221 wt.% in SBA-15(C)-2000, 227 wt.% in SBA-15(C)-1300 and 227 wt.% in SBA-15(C)-800 (wt.% after drug adsorption using 10^−1^ mol·dm^−3^ DIC solution). 

### 3.7. Drug Release Properties

In vitro drug release measurements were realized in two model media differing in pH, at pH = 7.4 for the simulated intravenous solution/small intestine environment and at pH = 2 for the simulated gastric juice. In addition, drug release was also assessed at two different temperatures: (i) At T = 37 °C, representing a normal human body temperature, the conformation of PEI molecules should partially or totally close the pores. (ii) At T = 42 °C, representing a fever temperature, the conformation of PEI molecules should partially or totally open the pores. The amount of DIC released from unmodified/modified samples was investigated using UV spectroscopy at different time intervals (0.5, 1.5, 3.5, 5.5, 7.5, 9.5 and 24 h). The actual diclofenac sodium concentrations during in vitro drug release experiments were calculated using the absorbance at two wavelengths depending on the pH value at *λ* = 287 nm at pH = 7.4 and *λ* = 285 nm at pH = 2. The loaded amounts of DIC (in mg/g od solid) determined by TG measurements (see Section 3.5) were used to calculate the weight percentage (wt.%) of the amount of drug released in selected time intervals: 191.1 mg for SBA-15(C)/DIC, 205.6 mg for SBA-15(C)-800/DIC, 178.7 mg for SBA-15(C)-1300/DIC and 141.8 mg for SBA-15(C)-2000/DIC (see Section 3.6). Figure 12 presents the time-dependencies of diclofenac sodium released quantity from the prepared materials under different temperature and pH conditions. Table 7 summarizes the amounts of released diclofenac sodium after 24 h. 

First, release experiments were performed on unmodified silica material SBA-15(C)/DIC (see Figure 12a), and obtained results were used as “standard” for comparison with modified materials. As can be seen from Figure 12a, almost no release is observed at acidic pH = 2 as DIC release in wt.% is less than 10%. Under acidic conditions (pH = 2), diclofenac sodium salt is protonated and transformed to diclofenac acid, which has extremely low solubility in water, and this was reflected in the amount of drug released. At pH = 2, only a small influence of temperature was observed: approximately 1.5 times more drug was released at T = 42 °C (8.1 wt.%) compared to T = 37 °C (5.3 wt.%). A similar effect of the temperature on the DIC release properties was observed at pH = 7.4. The temperature had only a small effect on the release profiles: the DIC release from the surface of SBA-15(C)/DIC was a little more continuous at T = 37 °C compared to T = 42 °C. The maximal amounts of the drug released after 24 h at pH = 7.4 were similar: 89.4 wt.% at 37 °C and 91.6 wt.% at 42 °C. As shown in Figure 12a, the pH of the medium plays an important role in the drug release process, mainly at low pH. 

The DIC release properties for PEI-modified samples were investigated at both pH levels and both temperatures. Results obtained at pH = 2 are very similar to those obtained for SBA-15(C)/DIC: only a small percentage of adsorbed DIC (<16%) was released at T = 37 °C and T = 42 °C. As can be seen from Figure 12b, at pH = 7.4 and both temperatures, the number of amine groups did not affect the total amount of drug released. The molecular weight/number of monomer units had no impact on the amount of released drug (a similar trend was found in TG analysis, as seen in Section 3.6), as DIC amounts released after 24 h were approximately the same: 51.1 wt.% at 37 °C and 73.5 wt.% at 42 °C for material SBA-15-800(C)/DIC, 48.1 wt.% at 37 °C and 75.1 wt.% at 42 °C for sample SBA-15-1300(C)/DIC and 49.2 wt.% at 37 °C and 72.7 wt.% at 42 °C for support SBA-15-2000(C)/DIC. The lower value of the maximum amounts of drug released found (close to 70 wt.%) compared to SBA-15(C)/DIC (close to 90 wt.%) can be explained by the creation of specific interactions (hydrogen bond formation and electrostatic interactions) between the drug molecules and primary/secondary amine functional groups. Furthermore, as can be seen from the above values, the temperature has an influence on the amount of drug released. Indeed, for all materials containing PEI molecules, the DIC release amount found at T = 42 °C is larger than the one at T = 37 °C, and the difference observed is between 22 and 27 wt.%, depending on materials. At T = 37 °C, the PEI molecules develop hydrogen bonding, and the entrances of the pores are partially blocked, whereas at T = 42 °C, the hydrogen bond system is temporarily disrupted and the pores are opened, favoring an important release of the drug. These results and DSC measurements (see Figure 9d) confirm the thermosensitivity of our carriers, even if the pore blocking is not total.

For a better interpretation of drug release results, the solubility of diclofenac sodium was studied under experimental conditions as described in the present work. The dispersed drug in the media was stirred for 24 h, and the dissolved quantity of drug was measured by UV-VIS spectroscopy. The calculated drug solubilities were 1399 μg·cm^−3^ at pH = 7.4 and T = 37 °C, 1508 μg·cm^−3^ at pH = 7.4 and T = 42 °C, 1.76 μg·cm^−3^ at pH = 2 and T = 37 °C and 1.91 μg·cm^−3^ at pH = 2 and T = 42 °C, and the determined results are in very good agreement with the published data [48,49]. As can be seen from obtained values, the temperature had a weak effect on the drug solubility. However, with increasing temperature, the amount of dissolved drug increased. The effect of temperature and pH on the amount of drug released can be demonstrated with the SBA-15(C)-2000/DIC sample and a pure drug. Three milligrams of support SBA-15(C)-2000/DIC (containing 0.43 mg of diclofenac sodium, based on TG) was used in the drug release studies from the carrier, which was dispersed in 50 cm^3^ of saline solution. For pure diclofenac sodium, complete dissolution of the drug would occur at pH = 7.4 (solubility of DIC is 69.95 mg·cm^−3^ at T = 37 °C and 75.40 mg·cm^−3^ at T = 42 °C in 50 cm^3^). At pH = 2, the amount of drug released would be 20.5 wt.% at T = 37 °C and 22.2 wt.% at T = 42 °C (*m*_100%_ = 0.43 mg). However, the SBA-15-2000/DIC sample showed a gradual release of the drug, and the maximum amounts determined after 24 h were lower (at pH = 7.4, 49.2 wt.% (T = 37 °C) and 72.7 wt.% (T = 42 °C); at pH = 2, 3.6 wt.% (T = 37 °C) and 10.7 wt.% (T = 42 °C)). This experimental result also points to the creation of intermolecular interactions between PEI and the drug molecules and the thermosensitivity of prepared DDSs. Moreover, lower amounts of DIC released from supports at pH = 2 when compared to pure diclofenac sodium could be an advantage of our prepared materials. It is known from clinical studies that one of the side effects of DIC with its long-term use or at high doses is the formation of gastric ulcers. It is estimated that NSAIDs, including DIC, cause around 2000 deaths a year due to the perforation of stomach ulcers [50,51]. SBA-15(C)-2000/DIC material suppresses the solubility of NSAIDs in gastric fluid (pH = 2) by 92% at T = 37 °C or 51% at T = 42 °C in comparison to the pure drug, which may suppress the risk of gastric ulcers.

Finally, the DIC release was also studied using commercial support, named Diclofenac Duo, from the company PharmaSwiss Figure 12. This product was reported to have a controlled release of the drug. A qualitative analysis was performed on a sample of the commercial product, which confirmed the presence of the undefined silicate component. The controlled release indicated that the material was unlikely to be porous, and thus, the drug was only granulated. The drug was released in less than 5 h, and the equilibrium drug concentrations equalized almost immediately. No effect of temperature was evidenced, as also found for the SBA-15(C) support.

To simulate the process of drug delivery through the digestive tract, an experiment with continuously changing pH and temperature was performed. After peroral administration, the carrier material passes through media with different pH values, starting from the stomach (pH = 2), moving through the large intestine (pH = 5.5) and, finally, entering the small intestine (pH = 7.4). 

The material SBA-15(C)-1300/DIC was selected and used for the mentioned experiment, and the effect of applied dynamic conditions on the amount of drug released is shown in Figure 13. Figure 13 represents a fragmented graph with different applied conditions. In the first part, at pH = 2 and a normal body temperature of T = 37 °C, a minimum drug release of 1.8 wt.% after 30 min under applied conditions was observed. After this time, the system was heated to T = 42 °C, and a higher amount of drug (2.9 wt.%) was released (second fragment). Applied conditions examined the behavior of the carrier in the stomach at normal and inflammatory temperatures. Subsequently, the pH of the system was increased to 5.5 at a constant temperature (T = 42 °C), which simulated the conditions in the large intestine during fever (the third part of Figure 13). As shown in Figure 13, more than 50% of the drug was released after 2.5 h. The fourth part shows the simulated conditions of the small intestine environment at pH = 7.4 and T = 42 °C, when the maximum release of diclofenac sodium occurs with a value of 73.4 wt.%. In the case of lower temperature (T = 37 °C) at the same pH, DIC is readsorbed and the maximum amount released decreases to 47.5 wt.% (after 1 h). The above-observed results prove that after the peroral administration of our designed DDS, a gradual release of the drug occurs that depends on the pH and temperature in the gastrointestinal tract. Higher drug release occurs with increasing pH and also at higher temperatures, which imitate the inflammatory state of the body.

### 3.8. Drug Release Kinetics

Various kinetic models (zero order, first order, Korsmeyer–Peppas, Hixson–Crowell and Higuchi) were applied to fit the data obtained from drug release studies to determine the DIC release rate and its release mechanism. For this purpose, the data obtained at pH = 7.4 were analyzed, as the amount of drug release at pH = 2 was negligible due to the low solubility of the diclofenac sodium in an acidic environment. Calculated release rate constants with corresponding *r^2^* values based on the linear regression analysis of selected models are summarized in Table 8. 

The release behavior of diclofenac sodium from sample SBA-15(C) at T = 37 °C and T = 42 °C and samples SBA-15(C)-800 and SBA-15(C)-1300 at T = 42 °C can be described by the Higuchi kinetic equation. The Higuchi model has commonly been used to model diffusion-controlled release processes of drug release from porous matrixes [52,53,54]. Higuchi’s model describes the drug release from support as the square root of a time-dependent process based on the Fickian diffusion process [55].

The Korsmeyer–Peppas model was applied for the samples with drug release of a slower rate, as only the first 60% of drug release data are supposed to be fitted in this model. From the calculated parameters, it was proven that the Korsmeyer–Peppas model is the best model for samples SBA-15(C)-800, SBA-15(C)-1300 and SBA-15(C)-2000 at T = 37 °C with regression coefficients (*r^2^*) of 0.9364, 0.9544 and 0.9932, respectively (see Table 8). The obtained diffusion exponent value (*n*) was >1 for all samples, suggesting that diclofenac sodium release follows “super case-II” transport. The mentioned drug release model is typically used in the analysis and fitting of drug release data related to polymeric systems in which the release mechanism is not well known or when the release involves more than one type of drug release phenomenon [56,57]. This analysis of the release data also indicated the thermosensitivity of the polymers located on the surface of supports. At T = 37 °C, when polymers are in a closed conformation, a slower drug release is observed compared to T = 42 °C (open conformation).

The best fit for the commercial product Diclofenac Duo was obtained with the Hixson–Crowell model with regression coefficients 0.9669 and 0.9502. The Hixson–Crowell cube root equation is a mathematical model applied and frequently used to describe powder dissolution or drug release from specific formulations. The shape of the drug dosage form is spherical, and the size will decrease as the system dissolves. For drug dosage forms, this model best describes release from erodible matrixes such as sustained-release tablets/granules [58].

## 4. Conclusions

In the present work, we prepared, characterized and described novel thermosensitive materials based on mesoporous silica SBA-15 functionalized with polyethylenimines for controlled drug release of diclofenac sodium salt. The as-synthesized SBA-15 matrix was prepared by the sol–gel method, and the surfactant located in pores was carefully removed by a combination of extraction and calcination processes. Subsequently, the surface of the calcined material was modified with chloropropyl groups using a grafting procedure. The last synthetic step in the preparation of supports consisted of chloropropyl groups’ condensation reaction and PEI molecules. In this study, PEI molecules present different molecular weights and, thus, different numbers of monomer units/numbers of amine groups. PEI polymers on the surface serve as thermosensitive molecules, which at lower temperatures form hydrogen bonds with each other and, due to their bulk, should block the entry into the pores. At higher temperatures, there is an initial disruption of intermolecular interactions, which should cause drug release. The mentioned conformational change was confirmed by DSC measurements. The successful synthesis of the prepared supports was confirmed by the results obtained using different analytical techniques such as infrared spectroscopy (IR), atomic force microscopy (AFM), small-angle X-ray diffraction (SXRD), nitrogen adsorption, transmission electron microscopy (TEM) and photon cross-correlation spectroscopy (PCCS). The nonsteroidal anti-inflammatory drug diclofenac sodium was encapsulated in the prepared materials. The amount of incorporated drug in the solid matrixes, PEI-modified/unmodified samples, was determined by thermogravimetric analysis. The drug adsorption determined by UV-VIS spectroscopy using a solution of diclofenac sodium at the maximum concentration shows that the prepared materials are able to store up to 23 times more drug than the weight of the carrier used. Subsequently, experiments were performed to assess drug release from the prepared materials into media with different pH values (simulated gastric fluid (pH = 2) and small intestine environment/simulated body fluid (pH = 7.4)) and temperatures (T = 37 °C representing normal body temperature and T = 42 °C simulating inflammatory temperature). From the obtained release curves, it could be concluded that in the case of pH, a higher amount of drug was liberated at pH = 7.4 compared to pH = 2 due to the drug solubility. In the case of temperature, a higher amount of DIC released was observed at a higher temperature, and the observed results confirmed thermosensitivity of the PEI-modified materials. Moreover, the drug release properties of prepared compounds were compared to a commercial product under the same experimental conditions. Various kinetic models were applied to fit the drug release data to study the mechanism of diclofenac sodium release. It was demonstrated that drug release at pH = 7.4 could be described by applying Higuchi and Korsmeyer–Peppas models.

## Figures and Tables

**Figure 1 materials-14-01880-f001:**
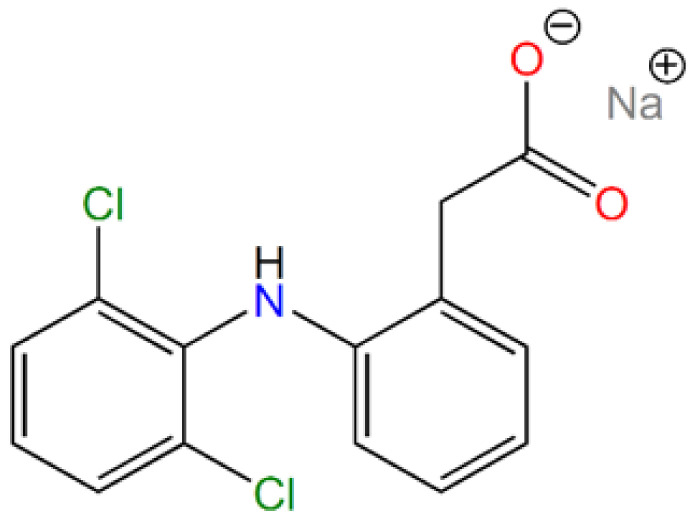
Molecular structure of diclofenac sodium salt.

**Figure 2 materials-14-01880-f002:**
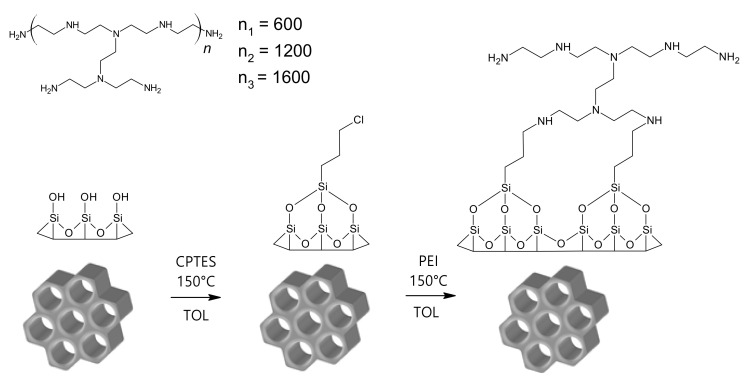
Scheme of grafting and surface modification processes.

**Figure 3 materials-14-01880-f003:**
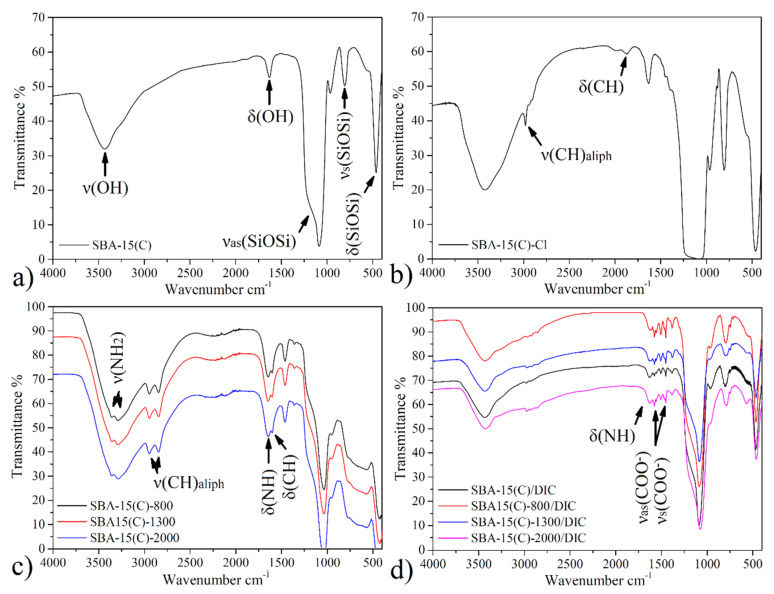
Infrared spectra of (**a**) SBA-15(C), (**b**) SBA-15(C)-Cl, (**c**) polyethyleneimine (PEI)-modified samples and (**d**) drug-loaded materials.

**Figure 4 materials-14-01880-f004:**
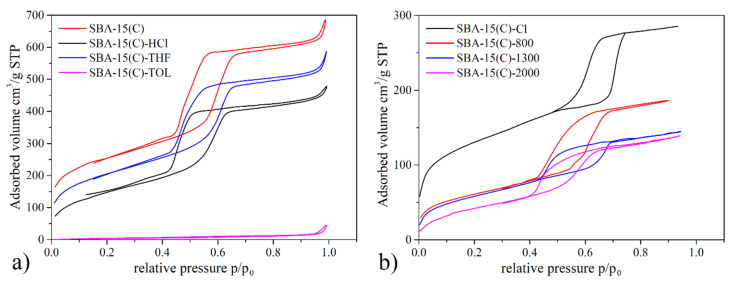
N_2_ adsorption/desorption isotherms measured at −196 °C of (**a**) the SBA-15(AS) during the surfactant removal process and (**b**) SBA-15(C) after chloropropyl grafting and PEI modification.

**Figure 5 materials-14-01880-f005:**
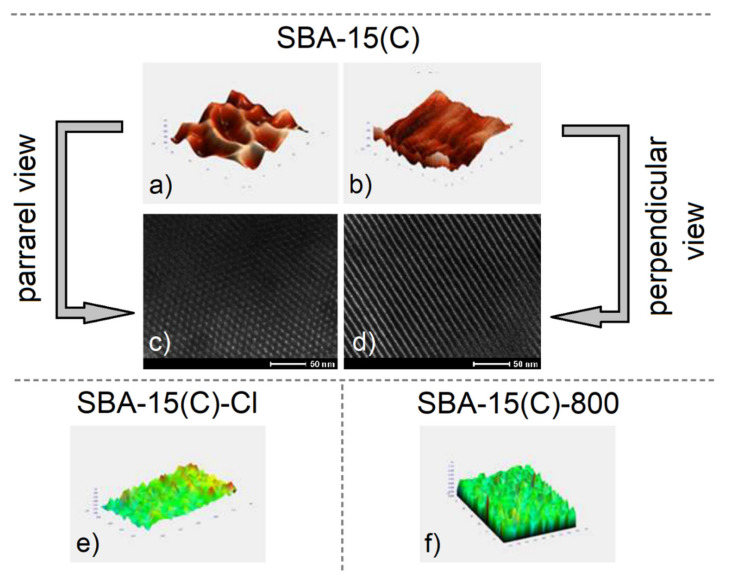
AFM scan images and TEM micrographs of calcined SBA-15(C) material showing hexagonal channels (**a**,**c**) parallel to the channels and (**b**,**d**) perpendicular to the pore direction. AFM images of (**e**) propylchloride-grafted material SBA-15(C)-Cl and (**f**) polyethylenimine-modified sample SBA-15(C)-800.

**Figure 6 materials-14-01880-f006:**
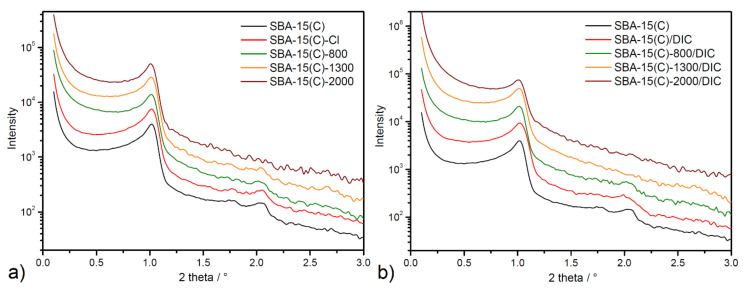
SA-XRD patterns of SBA-15 samples after (**a**) PEI grafting process and (**b**) diclofenac sodium encapsulation in logarithmic scale.

**Figure 7 materials-14-01880-f007:**
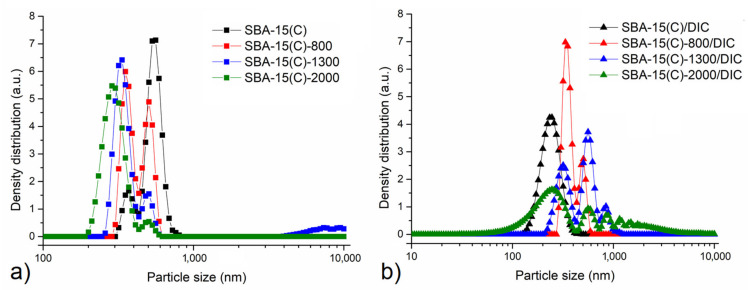
Particle size distribution of SBA-15 samples after (**a**) surface modification with PEI and (**b**) diclofenac sodium loading in logarithmic scale determined by photon cross-correlation spectroscopy.

**Figure 8 materials-14-01880-f008:**
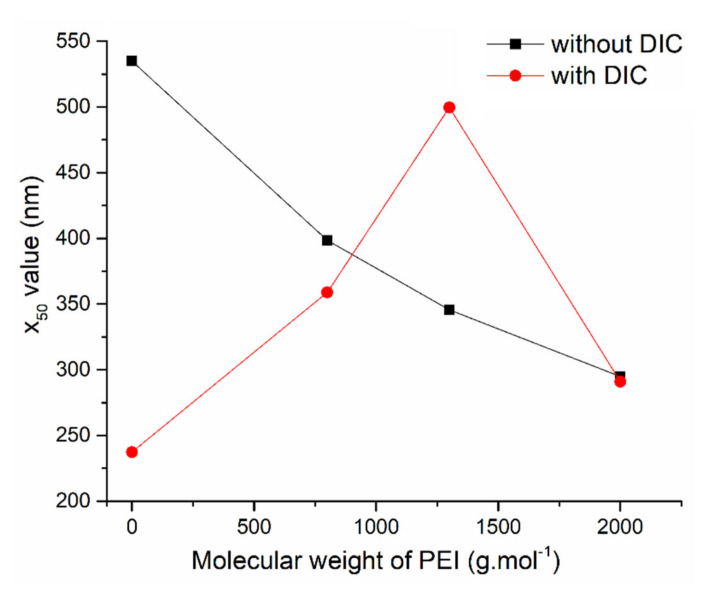
The relationship between average particle size x**_50_** and molecular weight of PEI after PEI surface modification (black curve) and diclofenac sodium loading (red curve).

**Figure 9 materials-14-01880-f009:**
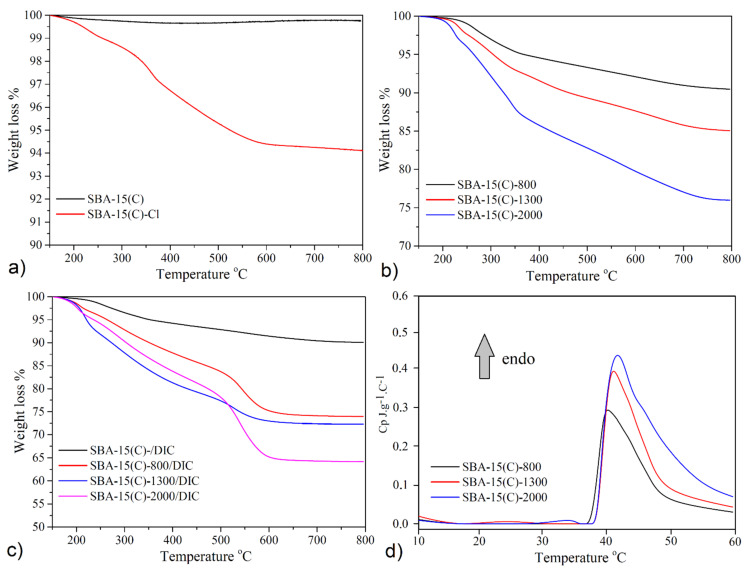
Thermogravimetric curves of (**a**) calcined (SBA-15(C)) and chloropropyl-grafted (SBA-15(C)-Cl) samples, (**b**) materials after PEI modification and (**c**) drug-loaded samples. (**d**) Results of DSC analysis of SBA-15(C)-800, SBA-15(C)-1300 and SBA-15(C)-2000.

**Figure 10 materials-14-01880-f010:**
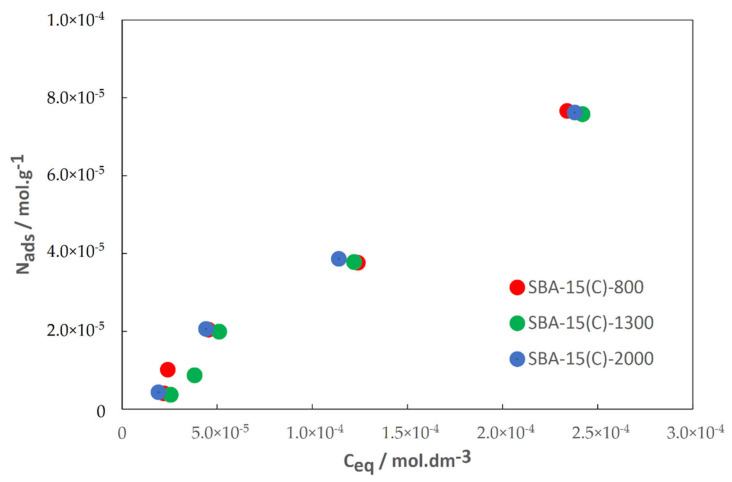
Diclofenac sodium adsorption isotherms on SBA-15(C)-800, SBA-15(C)-1300 and SBA-15(C)-2000.

**Figure 11 materials-14-01880-f011:**
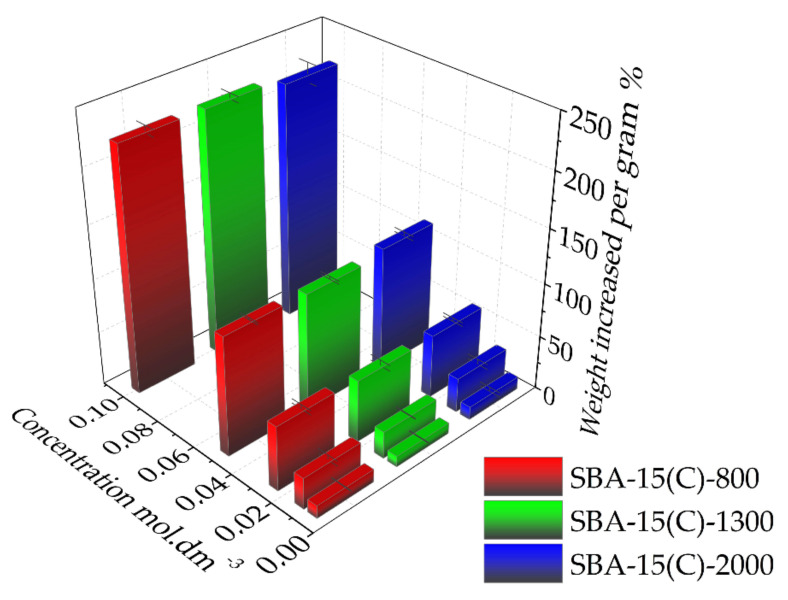
Amounts of loaded drug in the prepared PEI-modified silica samples with increasing concentration of diclofenac sodium solution.

**Figure 12 materials-14-01880-f012:**
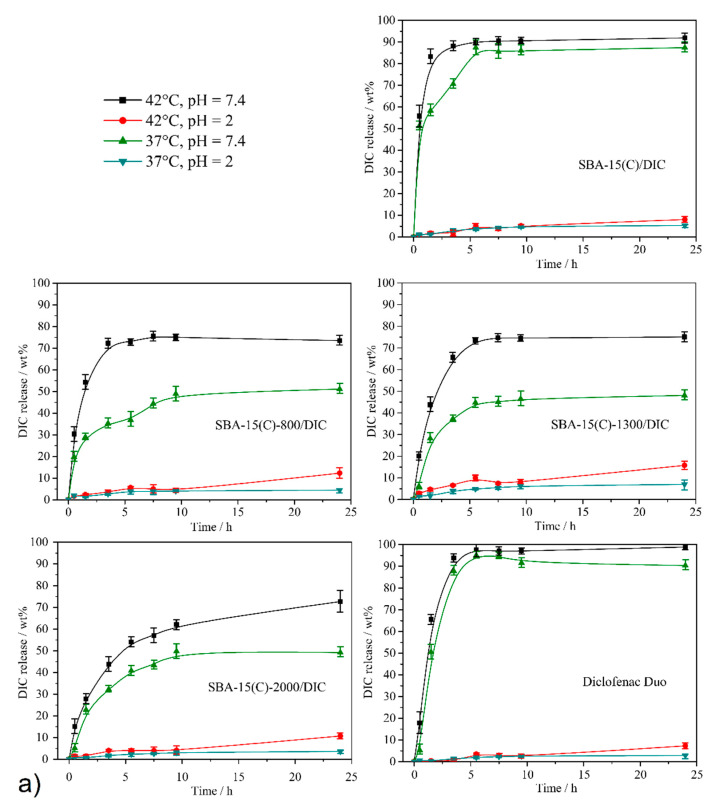
(**a**) Time-dependent release curves of diclofenac sodium salt from prepared materials (SBA-15(C), SBA-15(C)-800, SBA-15(C)-1300 and SBA-15(C)-2000) and Diclofenac Duo under different pH and temperature conditions. (**b**) Release curves of selected materials at pH = 7.4 and both temperatures for a clear representation of the thermosensitivity of PEI-modified samples.

**Figure 13 materials-14-01880-f013:**
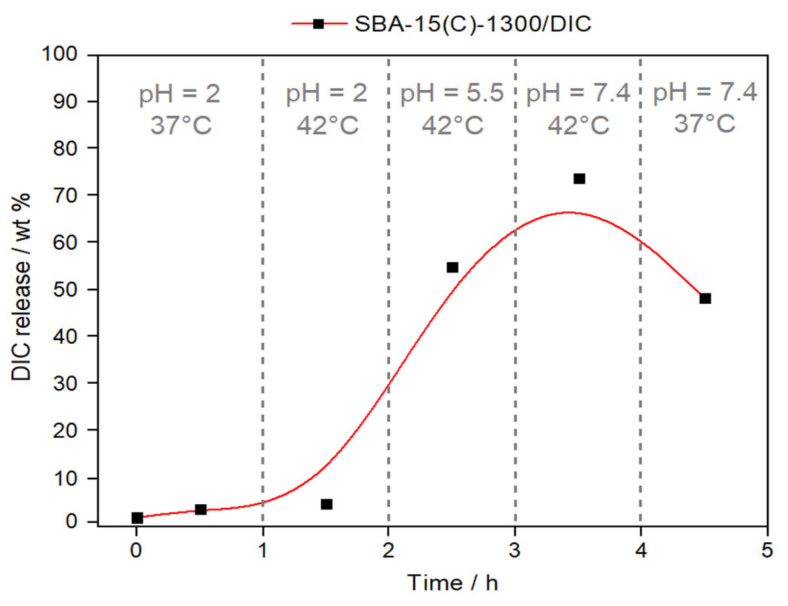
Release curve of diclofenac sodium from the sample SBA-15(C)-1300/DIC under dynamic conditions (continuous changes in temperature and pH).

**Table 1 materials-14-01880-t001:** Assignment of vibrations and corresponding wavenumbers to characteristic bands in IR spectra of surface-modified materials.

Sample	ν(OH)	ν(SiOSi)_s_	ν(SiOSi)_as_	δ(SiOSi)	δ(OH)	ν(CH)_aliph_	δ(CH)	δ(NH_2_)	ν(NH_2_)
SBA-15(C)	3437	809	1088	462	1638	-	-	-	-
SBA-15(C)-Cl	3427	800	1100	466	1638	2981	1453	-	-
						2931			
						2900			
SBA-15(C)-800	-	800	1100	475	1656	2954	1462	1602	3360
						2837			3300
SBA-15(C)-1300	-	800	1124	475	1638	2952	1462	1603	3283
						2837			3301
SBA-15(C)-2000	-	804	1098	475	1629	2953	1462	1602	3251
						2838			3300

ν—stretching; δ—scissoring; as—asymmetric; s—symmetric; ar—aromatic; aliph—aliphatic.

**Table 2 materials-14-01880-t002:** Assignment of vibrations and corresponding wavenumbers to characteristic bands in IR spectra of prepared materials with loaded drug.

Sample	ν(CH)_ar_	δ(CH)	ν(CCl)	ν(C=C)_ar_	ν(NH)	ν(COO^−^)_s_	ν(COO^−^)_as_	δ(NH)
SBA-15(C)/DIC	3080	1454	746	1555	3420	1450	1509	1611
				1580	3274			
SBA-15(C)-800/DIC	3082	1454	748	1555	3413	1451	1510	1609
				1580	3280			
SBA-15(C)-1300/DIC	3081	1453	746	1555	3421	1450	1511	1612
				1580	3274			
SBA-15(C)-2000/DIC	3080	1454	748	1555	3418	1453	1510	1610
				1580	3271			

**Table 3 materials-14-01880-t003:** The textural properties of prepared SBA-15 materials determined from N_2_ adsorption/desorption isotherms.

Sample	Surface Area	Pore Size	Pore Volume
	(m^2^.g^−1^)	(nm)	(cm^3^.g^−1^)
SBA-15(C)-TOL	14	-	0.21
SBA-15(C)-HCl	547	7.2	0.64
SBA-15(C)-THF	744	8.1	0.82
SBA-15(C)	814	8.6	0.86
SBA-15(C)-Cl	470	6.1	0.41
SBA-15(C)-800	206	5.9	0.28
SBA-15(C)-1300	184	5.6	0.24
SBA-15(C)-2000	144	5.5	0.24

**Table 4 materials-14-01880-t004:** Calculated cell parameters from SA-XRD patterns of prepared materials.

Sample	2*θ* of (100)/°	*a*/Å
SBA-15(C)	1.019	100.0
SBA-15(C)-Cl	1.017	100.2
SBA-15(C)-800	1.014	100.5
SBA-15(C)-1300	1.014	100.5
SBA-15(C)-2000	1.010	100.9
SBA-15(C)/DIC	1.018	100.1
SBA-15(C)-800/DIC	1.019	100.0
SBA-15(C)-1300/DIC	1.018	100.1
SBA-15(C)-2000/DIC	1.010	100.9

**Table 5 materials-14-01880-t005:** Obtained results and calculated amounts of PEI molecules from thermogravimetric analysis for surface-modified samples in different units.

Sample	Weight Loss ^a^ wt.%	Mass of Organic Part ^b^ mg·g^−1^	Amount of Organic Part ^c^ mmol·g^−1^
SBA-15(C)-Cl	6.01	58.8	-
SBA-15(C)-800	9.80	92.6	0.116
SBA-15(C)-1300	14.95	146.5	0.113
SBA-15(C)-2000	24.02	237.5	0.118

^a^—mass losses in wt.% on TG curves in the temperature range of 150–800 °C; ^b^—the mass of PEI molecules in mg on the surface of support per 1 g of material; ^c^—the amount of PEI molecules in mmol on the surface of support per 1 g of material.

**Table 6 materials-14-01880-t006:** Obtained results and calculated amounts of diclofenac sodium from thermogravimetric analysis for drug-loaded samples (see the legend under the table).

Sample	Total Weight Loss ^a^ wt.%	DIC Weight Loss ^b^ wt.%	DIC Mass ^c^ mg·g^−1^	DIC Mass Per Surface ^d^ mg·m^−2^
SBA-15(C)/DIC	16.17	15.95	191.1	0.235
SBA-15(C)-800/DIC	26.95	17.15	205.6	0.998
SBA-15(C)-1300/DIC	29.85	14.90	178.7	0.971
SBA-15(C)-2000/DIC	35.85	11.83	141.8	0.985

^a^—mass losses in wt.% on TG curves in the temperature range of 150–800 °C; ^b^—the mass loss corresponding to diclofenac sodium in wt.%; ^c^—the amount of diclofenac sodium in mg encapsulated in support per 1 g of material; ^d^—the amount of diclofenac sodium in mg encapsulated in support per m^2.^

**Table 7 materials-14-01880-t007:** Maximal amounts of drug released under different pH and temperature conditions after 24 h.

	pH and Temperature (°C) Conditions			
Sample	pH = 2		pH = 7.4	
	37 °C	42 °C	37 °C	42 °C
SBA-15(C)/DIC	5.3	8.1	89.4	91.6
SBA-15-800(C)/DIC	4.4	12.3	51.1	73.5
SBA-15-1300(C)/DIC	7.0	15.8	48.1	75.1
SBA-15-2000(C)/DIC	3.6	10.7	49.2	72.7
Diclofenac Duo	2.8	7.3	90.3	98.9

**Table 8 materials-14-01880-t008:** Fitting parameters of different kinetic models for the diclofenac sodium release amounts at pH = 7.4.

		Zero Order	First Order	Korsmeyer– Peppas	Higuchi	Hixson– Crowell
Sample	Temperature (°C)	*k*_0_ (mol.dm^−3^·h^−1^) *r^2^*	*k_1_*(h^−1^) *r^2^*	*k_KP_*(h^−n^) *n* *r^2^*	*k_H_*(h^−0.5^) *r^2^*	*k_HC_*(h^−1/3^) *r^2^*
SBA-15(C)	37	12.126 0.7064	0.6909 0.4863	*	21.594 0.9603	0.3442 0.8632
	42	12.437 0.5548	0.3732 0.7581	*	37.046 0.8144	0.3827 0.6893
SBA-15(C)-800	37	1.5813 0.5341	0.0239 0.6172	0.0003 2.7 0.9364	10.055 0.8262	0.0322 0.5897
	42	8.4882 0.6977	0.1747 0.8097	0.0025 1.7 0.8073	27.842 0.9064	0.2813 0.8217
SBA-15(C)-1300	37	1.6105 0.4397	0.0179 0.4459	0.0001 3.4 0.9544	8.7135 0.6318	0.0316 0.4696
	42	9.3551 0.8143	0.0490 0.4558	0.0118 1.4 0.8398	29.258 0.9618	0.0583 0.4352
SBA-15(C)-2000	37	1.7957 0.5268	0.0216 0.5467	0.0009 2.3 0.9932	11.4221 0.8156	0.0355 0.5604
	42	2.5615 0.6401	0.1027 0.9345	0.0043 1.8 0.8773	13.708 0.8788	0.0609 0.7425
Diclofenac Duo	37	18.435 0.8911	0.4511 0.9261	*	46.3081 0.9301	0.5726 0.9699
	42	3.0354 0.3521	0.1713 0.6017	*	21.2451 0.6599	0.6291 0.9502

* Because of the rapid drug release in the first hours, Korsmeyer–Peppas model could not be applied.

## Data Availability

The data presented in this study are available on request from the corresponding author.

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
