# Peer review of "Thermosensitive Drug Delivery System SBA-15-PEI for Controlled Release of Nonsteroidal Anti-Inflammatory Drug Diclofenac Sodium Salt: A Comparative Study"

_materials, 2021, doi:10.3390/ma14081880_

Round 1

Reviewer 1 Report

The study presented in this paper is interesting and significant for the scientific community especially in the field of drug delivery system.  all the methodologies are well explained and the results support the aim of the study.  I have few comments to be listed:

The quality of figures need to be improved for instance: Fig. 4 and Fig.10 are way too blurred. In other figures, captions (a), (b), etc. are not inlined with the fig. and can be prepared.

After addressing the above-mentioned issues, I recommended this manuscript to be published in Materials.

Author Response

Reviewer 1

The study presented in this paper is interesting and significant for the scientific community especially in the field of drug delivery system.  all the methodologies are well explained and the results support the aim of the study.  I have few comments to be listed:

  1. The quality of figures need to be improved for instance: Fig. 4 and Fig.10 are way too blurred. In other figures, captions (a), (b), etc. are not inlined with the fig. and can be prepared.

Answer: The quality and captions of all figures in the manuscript were improved and supplemented.

After addressing the above-mentioned issues, I recommended this manuscript to be published in Materials.

Reviewer 2 Report

  • There should be a space between words and a following reference.
  • After the first time that drug delivery system is abbreviated to DDS, it should be abbreviated throughout the rest of the manuscript. It is written out again, for instance, at the end of the introduction section.
  • Figure 5 should probably include just one of the three shown examples for a), b), and c). This way the images can be enlarged for clarity – I’m not sure much more information is provided by two additional images in each case.
  • I suggest a read through to eliminate unnecessary language and run-on sentences as well as adding commas and other small connector words (the’s or a’s) where needed.

Here are several wording suggestions I have for the manuscript:

Starting on line 43: “Anti-inflammatory medications such as analgesics and nonsteroidal anti-inflammatory drugs (NSAIDs)[2] are effective inflammatory suppressants. However, these medications are often invasive and addictive and can exhibit negative long-term side effects such as liver, kidney, heart, and bone marrow damage.”

DDS shouldn’t be followed by “carrier.” I assume the drug delivery system is a carrier itself and that you are not meaning a carrier for the drug delivery system itself.

Line 646, “characterized” should be followed by a comma.  

Line 666: “Drug adsorption was also determined in solution 665 by UV-VIS spectroscopy, showing that the carriers were able to store up to 230% of the drug.” 230 % should be clarified in the conclusion with more specific language. The DDS is not carrying more drug then it came in contact with…

Author Response

Reviewer 2

  1. There should be a space between words and a following reference.

Answer: The spacing between words and corresponding references was performed throughout the text of the manuscript.

  1. After the first time that drug delivery system is abbreviated to DDS, it should be abbreviated throughout the rest of the manuscript. It is written out again, for instance, at the end of the introduction section.

Answer: The phrase drug delivery system has been replaced by the abbreviation DDS in the manuscript.

  1. Figure 5 should probably include just one of the three shown examples for a), b), and c). This way the images can be enlarged for clarity – I’m not sure much more information is provided by two additional images in each case.

Answer: The number of figures in Fig. 5 was reduced according to reviewer´s suggestion (see page 13 in the manuscript).

  1. I suggest a read through to eliminate unnecessary language and run-on sentences as well as adding commas and other small connector words (the’s or a’s) where needed.

Here are several wording suggestions I have for the manuscript:

Starting on line 43: “Anti-inflammatory medications such as analgesics and nonsteroidal anti-inflammatory drugs (NSAIDs)[2] are effective inflammatory suppressants. However, these medications are often invasive and addictive and can exhibit negative long-term side effects such as liver, kidney, heart, and bone marrow damage.”

DDS shouldn’t be followed by “carrier.” I assume the drug delivery system is a carrier itself and that you are not meaning a carrier for the drug delivery system itself.

Line 646, “characterized” should be followed by a comma.  

Line 666: “Drug adsorption was also determined in solution 665 by UV-VIS spectroscopy, showing that the carriers were able to store up to 230% of the drug.” 230 % should be clarified in the conclusion with more specific language. The DDS is not carrying more drug then it came in contact with…

Answer: The text of the manuscript has been carefully read, the necessary dashes in the sentences have been added and the proposed additions have also been incorporated. Any further adjustments and design changes will be made in the publication process by journal Materials.

Reviewer 3 Report

The manuscript reports on the synthesis of mesoporous silica with polyethylenimine modified surface for diclofenac sodium (DIC) delivery. In vitro characterization showed that the drug release was faster at higher temperature and neutral pH. The authors claim that the carrier thus has thermosensitivity. The manuscript can only be considered for publication after revision with the supplementary of experimental data to address the following points.

  1. There is no evidence to prove that PEI is thermosensitive on SBA-15s, and hence led to the thermosensitivity of the drug release. Additional data are necessary to support the authors’ conclusion.
  2. The solubility of DIC should be demonstrated at different pHs and especially at different temperatures to support the authors’ statement that the formulation is stimuli responsive due to the features of the carrier.
  3. The morphology, particle size and dispersity of the SBA-15 materials should be examined.

Author Response

Reviewer 3

The manuscript reports on the synthesis of mesoporous silica with polyethylenimine modified surface for diclofenac sodium (DIC) delivery. In vitro characterization showed that the drug release was faster at higher temperature and neutral pH. The authors claim that the carrier thus has thermosensitivity. The manuscript can only be considered for publication after revision with the supplementary of experimental data to address the following points.

  1. There is no evidence to prove that PEI is thermosensitive on SBA-15s, and hence led to the thermosensitivity of the drug release. Additional data are necessary to support the authors’ conclusion.

Answer: The thermosensitivity of prepared materials was proven by differential scanning calorimetry measurements. Samples displayed endothermic peaks/effects with a maximum at 40.2, 41.1 and 41.7°C (see Fig. 9 on page 19). Corresponding results are described in section 3.5 Thermogravimetric analysis and calorimetry, please see page 19 in the manuscript.  

  1. The solubility of DIC should be demonstrated at different pHs and especially at different temperatures to support the authors’ statement that the formulation is stimuli responsive due to the features of the carrier.

Answer: The solubility of diclofenac sodium at different temperatures and pH was estimated by UV-VIS measurements. The detailed experimental procedure is described on page 6, obtained results and discussion about the comparison of prepared supports and drug solubility are described on page 23.

  1. The morphology, particle size and dispersity of the SBA-15 materials should be examined.

Answer: The morphology of prepared materials was investigated by AFM and small-angle X-ray diffraction in the original manuscript. According to the reviewer's suggestion, additional measurements of TEM and photon cross-correlation spectroscopy (PCCS) were performed for a more detailed study of morphology, particle size and dispersivity of materials. The obtained results are discussed on page 13 for TEM and pages 15-17 for PCCS.

Reviewer 4 Report

This manuscript prepared, characterized and described novel thermosensitive materials based on mesoporous silica SBA-15 functionalized with polyethylenimines for controlled drug release of diclofenac sodium salt. A higher amount of drug was released at pH = 7.4 compared to pH = 2 due to the drug solubility. In the case of temperature, a higher released amount of DIC was observed at a higher temperature. The data of the manuscript are enough to be published. However, the main drawback of the manuscript is the defective figure typesetting. I suggest it should be published after revision. Here are the questions about the manuscript.

  1. In figure 3c-d, you’d better divide infrared spectra curves for different samples to make a difference.
  2. Where is the figure 4a and 4b?
  3. You’d better put figure 5a-5c in just one page before publishing.
  4. The error bar should be added for drug loaded amount in figure 9.
  5. In figure 10, the error bar should be added for drug release curves.
  6. What’s the advantages of this drug delivery system? The authors should discuss them in introduction section.
  7. Some other studies (for example: Acta Biomaterialia, 2018, 72, 55-69; Carbohydrate Polymers 2017, 157, 493-502; Biomacromolecules, 2014, 15 (9), 3246-3252;) might be cited and discussed to improve the discussion of this work.

Author Response

Reviewer 4

This manuscript prepared, characterized and described novel thermosensitive materials based on mesoporous silica SBA-15 functionalized with polyethylenimines for controlled drug release of diclofenac sodium salt. A higher amount of drug was released at pH = 7.4 compared to pH = 2 due to the drug solubility. In the case of temperature, a higher released amount of DIC was observed at a higher temperature. The data of the manuscript are enough to be published. However, the main drawback of the manuscript is the defective figure typesetting. I suggest it should be published after revision. Here are the questions about the manuscript.

  1. In figure 3c-d, you’d better divide infrared spectra curves for different samples to make a difference.

Answer: The IR spectra of individual samples were divided according to the referee´s comment (please see Fig. 3 on page 10).

  1. Where is the figure 4a and 4b?

Answer: Figure 4 was modified and corresponding abbreviation 4a and 4b are more visible.

  1. You’d better put figure 5a-5c in just one page before publishing.

Answer: Figure 5 has been completely modified (please see page 13).

  1. The error bar should be added for drug loaded amount in figure 9.
  2. In figure 10, the error bar should be added for drug release curves.

Answer: The errors bars were included in Fig. 9 and Fig. 10 according to reviewer´s suggestion (please see page 21 and 24).

  1. What’s the advantages of this drug delivery system? The authors should discuss them in introduction section.

Answer: The introduction of the article was extended by a discussion concerning an overview of the applications of PEI polymers and their derivatives as DDS. The advantage and uniqueness of the presented study lie in the investigation of thermosensitive properties of PEI polymers, which were first studied on the porous silica material SBA-15.

  1. Some other studies (for example: Acta Biomaterialia, 2018, 72, 55-69; Carbohydrate Polymers 2017, 157, 493-502; Biomacromolecules, 2014, 15 (9), 3246-3252;) might be cited and discussed to improve the discussion of this work.

Answer: The proposed studies were incorporated into the introduction and also added to the literature.

Round 2

Reviewer 3 Report

The manuscript has been properly revised by the authors, and therefore has the condition for publication in Materials.

Author Response

We appreciate the time you spent reviewing the article and for your valuable comments.

Reviewer 4 Report

The paper can be accepted in its current form.

Author Response

(The authors gave the same response as above.)
